

# Analysis of geomagnetic observatory data and detection of geomagnetic jerks with the MOSFiT software package

Marcos Vinicius da Silva[1], Katia J. Pinheiro[1], Achim Ohlert[2], and Jürgen Matzka[2]

[1]Geophysics Department, Observatório Nacional - MCTIC, Rio de Janeiro, Brazil
[2]GFZ German Research Centre for Geosciences, Potsdam, Germany.

**Correspondence:** Matzka, J. (jmat@gfz-potsdam.de)

**Abstract.** MOSFiT (Magnetic Observatories and Stations Filtering Tool) is a python package to visualize and filter data from magnetic observatories and magnetometer stations. The purpose of MOSFiT is to automatically isolate and analyze the secular variation (SV) information measured by geomagnetic observatories data. External field contributions may be reduced by selecting data according to local time and geomagnetic indices and by subtracting the magnetospheric field predictions of the

CHAOS-7 model. MOSFiT calculates the SV by annual differences of monthly means and geomagnetic jerk occurrence time and amplitude are automatically calculated by fitting two straight-line segments in a user-defined time interval of the SV time series. Here, we present the new python package, validate it against independent results from previous publications, and show its application. In particular, we quantify the RMS misfit between SV derived from processing schemes and the SV predicted by CHAOS-7. Analysing the INTERMAGNET quasi-definitive data with MOSFIT allows for a timely investigation of SV

such as the detection of recent geomagnetic jerks. It can also be used for data selection for e.g., external field studies as well as for quality control of geomagnetic observatory data.

## 1 Introduction

The geomagnetic field results from a superposition of fields from internal and external sources, varying in a wide range of timescales from shorter than a second to longer than millions of years. The core field is produced by electric currents driven by

convective flow in the molten outer core and its intensity at the Earth's surface variesranges from 22.000 nT in South America to 67000 nT in south of Australia. Its time change, calculated as the first time derivative of the geomagnetic field, is called secular variation (SV) (e.g. (Bloxham et al., 1989)). Abrupt changes in the SV are called geomagnetic jerks (Courtillot et al., 1978; Bloxham et al., 2002). The other internal sources are the crustal and induced fields, generated by magnetised rocks in the lithosphere and by induced currents in the conductive mantle, respectively. The external field is produced by electric currents

in the ionosphere and magnetosphere and are usually a few tens of nT during magnetically quiet days and may reach more than 1000 nT in some regions of the Earth during disturbed days (Kono, 2010).

Geomagnetic observatories are fixed locations at the Earth's surface that measure continuously the geomagnetic field. Monthly and yearly mean data are widely used for core field studies (Matzka et al., 2010). However, these monthly and yearly means are not totally free from external field influences. Removing this remaining external field is a challenge when





isolating the SV. Different methods are applied, such as data selection based on geomagnetic indices, like $Kp$ (or Ap) and Dst
(Kotzé, 2011, 2017; Chulliat and Maus, 2014), or on the local time, such as nighttime selection in Chulliat and Maus (2014).
In addition, external fields predicted from geomagnetic field models can be subtracted from the data (Macmillan and Olsen,
2013; Finlay et al., 2020).

The International Real-time Magnetic Observatory Network (INTERMAGNET) establishes the standard for observatory
equipment and specifications for data quality. INTERMAGNET distributes software and a number of packages (https://
intermagnet.github.io/software.html) for data processing, calibration and quality control (e.g. MagPY and ObsMat), data vi-
sualization (e.g. Autoplot) and data check (e.g. check1min and DataCheck1S). Cox et al. (2018) developed an open-source
Python package (MagPySV) to process and filter observatory data, specifically in order to obtain SV. They developed a princi-
pal component analysis (PCA) method to denoise observatory data, based on Wardinski and Holme (2011), Brown et al. (2013)
and Feng et al. (2018). This package has options to denoise, process and correct baseline jumps from the hourly mean data,
downloaded from the World Data Center (WDC, Edinburgh). The WDC is a repository that includes digital definitive data
from INTERMAGNET and from other geomagnetic observatories (https://wdc.bgs.ac.uk/data.html), INTERMAGNET defini-
tive data are produced for one calendar year at a time and hence are published much later than the data acquisition. In the last
years, there is an increase need for close-to final observatory data shortly after their acquisition, for example, for the detection
of recent geomagnetic jerks (Pavón-Carrasco et al., 2021) as well as for global geomagnetic field modelling (Peltier and Chul-
liat, 2010). For this reason, INTERMAGNET established a new data type called quasi-definitive, which is published within
three-months of acquisition and for each observatory 98% of the quasi-definitive data must be within 5 nT of the definitive
data.

In this paper, we present a new python tool (MOSFiT) for isolating SV and detecting geomagnetic jerks, which automatically
download also quasi-definitive INTERMAGNET data. This may help to identify recent geomagnetic jerks, such as 2019-2020
event (Pavón-Carrasco et al., 2021), in a timely fashion.

In this paper, MOSFiT package and its potential applications are described. In addition, examples for MOSFiT validation
and jerk detection are presented. The package can also be used for quality control of geomagnetic observatory data, in a similar
fashion as done by British Geological Survey based on the methods described in Macmillan and Olsen (2013)

## 2   Setting up the package - Using MOSFiT

A guide of how to set up the package is given at GitHub (https://github.com/marcosv9/MOSFiT-package) as well as in the
supplementary material. Figure 1 shows a flowchart illustrating a typical sequence of the most important data processing
options. The processing according to this flowchart is already implemented in a dedicated function (sv_obs). However, the user
can combine any of the processing steps in any user-defined order. The diamond-shapes represent the interactions in the code,
in which the user needs to choose the options to process the data, the red rectangles represent the automatic calculations and
the green rectangles are the output (figures, files and statistics).





The first interaction in this sequence is the Hampel filter used for outlier detection and removal, see section 3.1 for a detailed description. Note that for the hampel, the data is resampled to hourly mean data in order to reduce computacional cost. The second interaction is the reduction of the external field according to local time or geomagnetic indices (3.2). The third step (Fig. 1) is the subtraction of the magnetospheric fields predicted by CHAOS-7 (3.3). The data can be visualized by using different means (minute, hourly, daily, monthly, annual as well as unfiltered minute means). Plots are always displayed on the screen, allowing the user to evaluate the results of the processing steps and also to select a suitable start and end time for the jerk detection. The user can choose if these plots and files should be saved or not. A specific directory is automatically created to store the saved files/plots. The outputs are separated by observatories, and all the data files contain a header with the observatory information and the chosen processing options. Then, the SV is calculated, and visualized by monthly means. The last interaction is the geomagnetic jerk detection, based on fitting two linear segments to the SV in a user-defined time window. The output is geomagnetic jerk occurrence time ($t_0$), geomagnetic jerk amplitude ($A$), and the coefficient of determination $R^2$ of the linear segments (see section 3.6).

## 3 Description of Methods

### 3.1 Outlier detection - Hampel filter

The Hampel filter is a robust outlier detector for time series, see Hampel (1974) and Pearson et al. (2016) for a detailed explanation. It is based on the median absolute deviation (MAD), since the MAD is less affected by outliers than the mean. Equation 1 is the $M$, where $xi$ is the ith observed data point and m the median for each window.

$$M = median(|x_i - m|) \tag{1}$$

It works like a sliding window over the time series. The filter sensitivity depends on the window size (number of samples) and the threshold (in multiples of standard deviation). Any data point exceeding the threshold is flagged as an outlier and replaced by the window median value. This approach follows Cox et al. (2018) way of denoising the WDC data. By default, MOSFiT takes a window of hundred hours and a threshold of 3 times the standard deviation. This choice is based on trial and error and, according to our visual inspection of SV time series, it works well for most observatories. But for some observatories other value combinations might work better.

### 3.2 External field reduction by data selection

One method to mitigate external field contributions is by selecting periods of low geomagnetic activity (Kauristie et al., 2017). In MOSFiT you can select periods of low $Kp$ index, or the international quiet days, or exclude data from the international disturbed days. There is often a trade-off between stringent criteria retaining only a small amount of data and less stringent criteria that keep more external field affects in the data. Another method is to select nighttime, as the E-region ionospheric





**Figure 1.** Flowchart of MOSFiT processing example as implemented in the interactive function sv_obs. The diamond shapes are the data processing sequence interactions, where the user chooses the options of data processing. Diamonds, red boxes, and green box represents user selections, processing steps, and output, respectively.

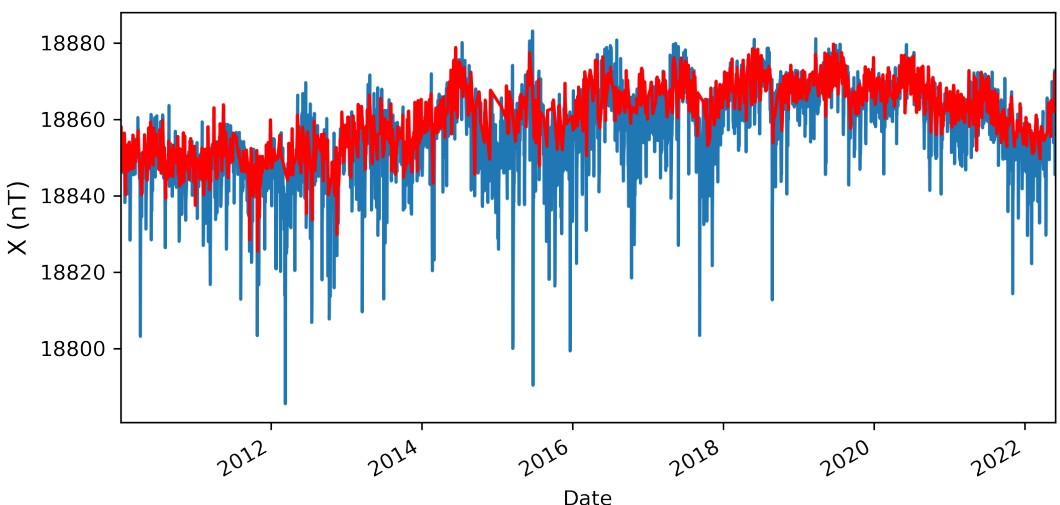

**Figure 2.** North geomagnetic field component (X) from Vassouras geomagnetic observatory (VSS). Daily means for all days (blue curve) and daily means for days with $Kp$ index $\leq 2$ calculated by the MOSFiT package (red curve).

conductivity is then very low. Data selection is not only useful for studying the SV, but also for ionospheric studies (Yamazaki and Maute, 2017) and internal field modeling (Kauristie et al., 2017).

### 3.2.1   $Kp$ **index**

The $Kp$ is a three-hour range index that is widely applied to data selection in geomagnetic field modelling and for studies of the ionosphere and magnetosphere (Kauristie et al., 2017). It is distributed by GFZ-Potsdam (https://kp.gfz-potsdam.de/en/), see Matzka et al. (2021) for a recent description of $K$ and $Kp$ index derivation. Selecting periods with $Kp \leq 2$ (default in MOSFiT) retains on average 30% of the available data (Yamazaki and Maute, 2017), but that depends on the position in the solar cycle and during periods of high solar activity, the $Kp$ limit might have to be increased in order to retain a sufficient number of data. The function automatically downloads updated $Kp$ index values from the GFZ website when the function is used. Figure 2 illustrates the MOSFiT $Kp$ index selection function for $Kp \leq 2$ applied to Vassouras (VSS) data from January 2010 to May 2022.

### 3.2.2   **International quiet and disturbed days**

The International quiet and disturbed days (IQDs and IDDs) are derived from the $Kp$ index and distributed by the GFZ-Potsdam, on a monthly basis on https://www.gfz-potsdam.de/en/Kp-index. MOSFiT provides options to select data based on



IDDs or IQDs, which completely remove the top 10 IDDs and keep the top 5 IQDs for each month of the dataset. Similar to
the $Kp$ index, the IDDs and IQDs lists are updated automatically when accessed by the package.

### 3.2.3   Nighttime selection

Nighttime geomagnetic data is less affected by overhead ionospheric E-region currents like, e.g. the (daytime) Sq current sys-
tem. This is due to the low E-region conductivity at night time. However, depending on the latitude of the station, other current

systems like the substorm current wedge can affect the data around local midnight at e.g. sub-auroral stations (McPherron
et al., 1973; McPherron and Chu, 2017). The default value for nighttime selection in MOSFiT is from 10 pm to 02 am local
time.

### 3.3   CHAOS-7 model correction

The CHAOS-7 geomagnetic field model (Finlay et al., 2020) is derived from Swarm, CHAMP, Orsted and SAC-C satellite and

ground observatory data. It predicts the contributions from the following sources: core, crust and magnetosphere. The MOSFiT
package includes the version CHAOS-7.10, spanning from 1999 until August, 2022. [1].

The inputs are geodetic longitude and colatitude, distance to the Earth's center and timestamps for which the field will be
predicted, and the RC index (see below). The model output is the radial, colatitude and azimuthal component ($B_r$, $B_\theta$ and $B_\phi$,
respectively) for each geomagnetic field sources, converted to the local XYZ-coordinate system that is used for geomagnetic

observatories.

The time-dependent core field is calculated up to spherical harmonic degree 20 and the static crustal field is calculated up
to degree 110. Magnetospheric fields are calculated in the geocentric solar magnetospheric (GSM) and solar magnetic (SM)
coordinate systems, both up to degree 2 and with hourly resolution. They can be can be subtracted from the observatory hourly
mean data in order to reduce external field influence. In order to calculate the SM contributions MOSFiT is automatically

updating the ring current RC index from http://www.spacecenter.dk/files/magnetic-models/RC/current/.

Figure 3 compares the SV calculated for Kakioka geomagnetic observatory (KAK) before and after subtraction of the
CHAOS-7 magnetospheric (GSM and SM) contribution. Additionally, Figure 3 presents the SV calculated from the time-
dependent CHAOS-7 core field prediction. Such comparison demonstrates the better characterization of the SV after the fil-
tering, explained by the good agreement with the predicted SV. Figure 3 shows that mostly the X component SV is influenced

by the magnetospheric field and that its reduction by the CHAOS model significantly improves the SV information in the
observatory timeseries.

---

[1]see http://www.spacecenter.dk/files/magnetic-models/CHAOS-7 for CHAOS release information



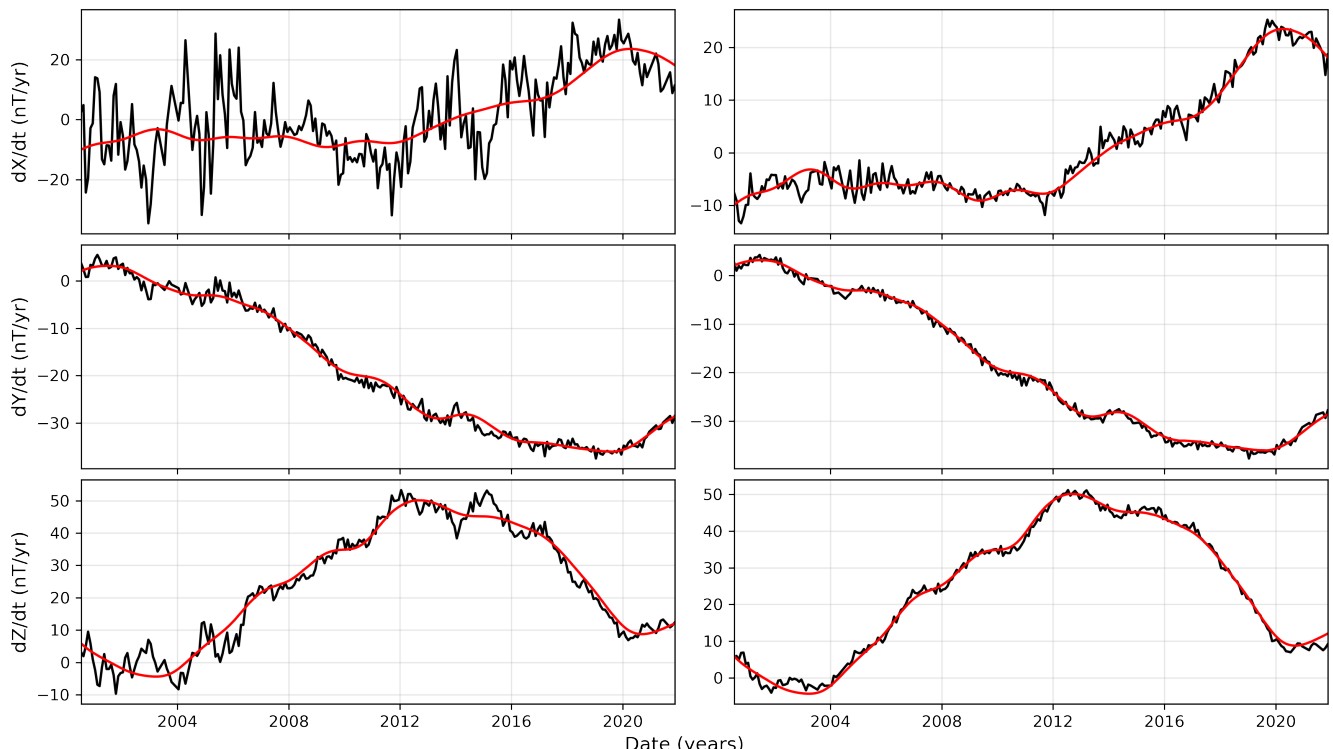

**Figure 3.** Kakioka magnetic observatory (KAK) secular variation (SV, black curves) calculated by MOSFiT before (left panels) and after (right panels) subtraction of magnetospheric field as predicted by CHAOS. SV calculated from CHAOS core field predictions are shown as red lines.

## 3.4 Data resampling

The package can resample geomagnetic data as hourly, daily, monthly and annual means that are centered to the middle of the time interval. By default MOSFiT calculates a mean value if at least 90% of data is available (St-Louis et al., 2020). For
example, to resample from minute means to hourly means, it is necessary to have a minimum of 54 minutes means within this UT hour. The default can be disabled by the user, when the observatory has low data availability or to avoid cascading effects, for example if a few missing minute mean values cause a missing hourly mean values which in turn cause some missing daily mean values and so on.

## 3.5 Secular variation calculation

The MOSFiT default method to calculate the SV at a time t is the monthly mean differences between t+6 months and t-6 months (Chulliat et al., 2010; Feng et al., 2018), well known as Annual Differences of Monthly Means (ADMM). Equation 2 represents the SV, where $t$ is time in months and $B$ is the magnetic field of any geomagnetic component. In most of the cases, monthly means are calculated from minute means (IAGA-2002, definitive and quasi-definitive data). Monthly means





are a better option than annual means for SV investigations as they allow to identify abrupt SV changes into more detail.

Importantly, this approach also reduces external field contributions by filtering out the seasonal variation from magnetospheric and ionospheric currents. Note that the solar quiet ionospheric currents are not modelled by CHAOS-7 and could otherwise introduce strong seasonal artifacts in the SV record (Mandea et al., 2000).

$$\frac{dB}{dt} = B_{(t+6)} - B_{(t-6)} \qquad (2)$$

### 3.6   Geomagnetic jerk detection

Geomagnetic jerks can be viewed as a sudden change in the first time derivative (SV) trend, characterized by a V-shaped pattern in the SV, or an abrupt change in the second time derivative (secular acceleration, SA) of the magnetic field. Usually, they are not observed simultaneously around the globe, i.e. the same event is observed shifted by several months at different observatories (Courtillot et al., 1978; Bloxham et al., 2002; Pinheiro et al., 2019).

Different methods to detect geomagnetic jerks have been applied in the literature and Brown et al. (2013) give an overview

of detected events, used data, and detection techniques, of which the most common were least square fitting of two straight lines (Le Mouël et al., 1982; Le Huy et al., 1998; Pinheiro et al., 2011), wavelet analysis (Alexandrescu et al., 1996) and visual detection (Mandea et al., 2000; Olsen and Mandea, 2007; Chulliat et al., 2010).

MOSFiT provides an automatic fitting of two straight lines segments by least squares in a user-specified time window. This requires that the user chooses a suitable time interval that contains a jerk surrounded by two linear segments. The package

makes use of PWLF (Jekel and Venter, 2019), a python library that fits continuous piecewise linear functions to one dimensional (1D) dependent variables. In our case, we want to fit two linear segments to the data without prescribing their intersection point, which is the geomagnetic jerk occurrence time ($t_0$). Global optimization is used to find the best intersection ($t_0$), by minimizing the sum-of-square error of the residuals. The detection is applied to each of the X, Y and Z component individually. The PWLF also gives the coefficient of determination ($R^2$) and the slopes of the first and second segments. The jerk amplitude ($A$) is the

slope of the second segment minus the slope of the first segment (Le Mouël et al., 1982; Pinheiro et al., 2011).

Figure 4 demonstrates the detection of the geomagnetic jerk in 2014 at Niemegk geomagnetic observatory (NGK) corrected by the CHAOS-7 magnetospheric field. The detection algorithm was applied to a time window from June 2012 to January 2018. The statistics about the detection are listed in Table 1. As expected, the SV in X has the largest misfit ($R^2 = 0.74$) since it is the most affected by the external field contribution. The SV in Y and Z show a very good fit ($R^2 = 0.97$ and 0.99, respectively).

The occurrence time $t_0$ was identical for the X and Z component (2014.62), and about 4 months earlier in the Y component (2014.27). The user should be aware that the chosen detection window may affect the jerk characterization and should only select windows that contain visually clear changes of trend in the SV. Otherwise, the automatically determined jerk occurrence time and amplitude can be less accurate or wrong. The algorithm always fits two linear segments into the time chosen window, even if no apparent jerk signs exist. It is a user's task to interpret the results by, for example, setting a criteria to exclude jerks

with lower $R^2$ values.



**Table 1.** Statistics of MOSFiT geomagnetic jerk detection for the jerk in 2014 at Niemegk magnetic observatory secular variation. The parameters $R^2$, $A$ and $t_0$ are the coefficient of determination, jerk amplitude and occurrence time, respectively.

| SV | $R^2$ | $A$ (nT/yr) | $t_0$ (time) |
|----|-------|-------------|--------------|
| X | 0,74 | -1,73 | 2014,62 |
| Y | 0,97 | 7,25 | 2014,27 |
| Z | 0,99 | 4,43 | 2014,62 |



**Figure 4.** Niemegk geomagnetic observatory (NGK) SV. Dark blue, green and black are the SV for X, Y and Z geomagnetic components, respectively. Light blue lines give the SV from the CHAOS-7 predicted core field. The red lines show the linear segments identified by MOSFiT and indicate the start and end time of the user-selected detection window (June 2012 to January 2018).



# 4 MOSFiT package validation

## 4.1 Case Study 1: Global misfit with and without magnetospheric correction

Finlay et al. (2020) calculated the RMS misfit between the SV of CHAOS-7 magnetospheric corrected data of 181 magnetic observatories and the SV determined from the CHAOS core field. Their mean misfit for $B_r$, $B_\theta$ and $B_\phi$ is given in table 2. We denote this method as $RMSe_1$ and repeat this exercise with MOSFiT on data from 115 INTERMAGNET observatories from January 2000 until June 2022. Our results (also in Table 2) are very similar though not identical as there are some differences in the observatories, time interval and CHAOS model version. They indicate that the CHAOS implementation in MOSFiT works properly. In Figure 5 we show the global distribution of this RMS misfit for the individual observatories. As expected, largest misfits are observed at high latitudes, where there are the strongest unmodelled external field variations. The smallest misfits are observed in Europe, likely because here the density of geomagnetic observatories is the highest.

**Table 2.** Validation of the MOSFiT implementation for external field correction by CHAOS-7, against results from Finlay et al. (2020)

|  | MOSFit ($RMSe_1$) | Finlay et al. (2020) ($RMSe_1$) |
|---|---|---|
| $dB_r/dt$ | 3,73 | 3,73 |
| $dB_\theta/dt$ | 2,9 | 3,59 |
| $dB_\phi/dt$ | 3,5 | 3,31 |

In addition, we calculate the RMS difference between SV determined from uncorrected geomagnetic observatory data and SV determined from the CHAOS-7.11 core field. We denote this method as $RMSe_2$ and show global mean values for high, mid and low latitudes in table 3. It also shows the corresponding results for $RMSe_1$ and the percentage of improvement from $RMSe_1$ to $RMSe_2$.

Most improvement is seen for dX/dt at mid and low latitudes as the magnetospheric signal, which is modelled by CHAOS-7 and then subtracted, is strongest in the X component. At high latitudes the improvement is only around 30% since signals from the auroral electrojet and field aligned currents are not modelled by CHAOS-7. The east component is least affected by external fields and shows the lowest RMS values as well as the smallest improvements.

## 4.2 Case Study 2: Detection of the geomagnetic jerks in 2007, 2011 and 2014

In order to validate the MOSFiT methods, we automatically detected the jerks of 2007, 2011 and 2014 using observatory data from NGK, EBR, TAM and ASC. For the sake of comparison, we use methods similar to those described by Torta et al. (2015), i.e. only Y component without data selection or subtraction of external fields. Like Torta et al. (2015), we use only the data until latest the end of 2014. In Table 4 we compare MOSFiT results for occurrence time ($t_0$) and amplitude ($A$) with that of Torta et al. (2015). Additionally, we give the coefficient of determination ($R^2$) and the user specified start year/month and end year/month for the detection window. The detection window was selected visually from the SV plot, based on the





**Figure 5.** INTERMAGNET observatories RMSe between CHAOS-7 model internal field prediction SV and observed (filtered) SV. Maps of X, Y and Z SV from the top to the bottom, respectively. These results were produced by using MOSFiT package.



**Table 3.** CHAOS-7 model external field filtering effectiveness. Percentage of change between $RMSe_2$ and $RMSe^1$ (see text) for low, mid and high latitudes, for the SV of the X, Y and Z component.

|  | Mid Latitude | | | Low Latitude | | | High Latitude | | |
|---|---|---|---|---|---|---|---|---|---|
|  | $RMSe_2$ | $RMSe_1$ | % | $RMSe_2$ | $RMSe_1$ | % | $RMSe_2$ | $RMSe_2$ | % |
| dX/dt | 7,99 | 2,62 | 204,96% | 11,46 | 3,9 | 193,85% | 8,44 | 6,43 | 31,26% |
| dY/dt | 2,68 | 2,1 | 27,62% | 3,73 | 3,42 | 9,06% | 4,84 | 4,36 | 11,01% |
| dZ/dt | 5,36 | 2,5 | 114,40% | 4,04 | 3,27 | 23,55% | 9,65 | 6,45 | 49,61% |

characteristic V-shape and then repeated until the best misfit was achieved. Figure 6 shows the SV (blue dots), CHAOS-7 model SV prediction (green curves) and jerk detection (red lines) results of MOSFiT package.

The largest occurrence time difference between MOSFiT and Torta et al. (2015) for the jerk in 2007 is 0.15 years or 55 days for NGK, while the amplitudes are very similar. For the jerk in 2011 occurrence time reported by Torta for ASC is 2012.0, however Fig. 6 shows that both the observatory data and the CHAOS model prediction show a jerk around 2010, which is the same date as detected by MOSFiT. Jerk amplitudes detected by MOSFiT are very similar to those published by Torta et al. (2015). Finally, for the jerk 2014 the results shows a very good agreement for the occurrence time of the jerk as well as a good agreement for the amplitude at all four observatories.



**Figure 6.** Geomagnetic jerk detection of the 2007, 2011 and 2014 events, using the MOSFiT automatic method, for NGK, EBR, TAM and ASC. Blue dots are the non-filtered Y component SV, green lines are the CHAOS-7 model internal field SV prediction and the orange straight lines are the MOSFiT automatic jerk detection.





**Table 4.** MOSFiT detection of geomagnetic jerks in 2007, 2011 and 2014 at NGK, EBR, TAM and ASC observatories. Occurrence time ($t_0$) and amplitude ($A$) are shown for MOSFiT and for Torta et al. (2015) in parentheses. The coefficient of determination $R^2$ and the start and end year/month of the detection window are also shown, for this study.

| Jerk 2007 | | | | | |
|---|---|---|---|---|---|
| IMO | $t_0$ (yr) | $A$ (nT/yr) | $R^2$ | initial window | final window |
| NGK | 2005.85 (2006.0) | 4,75 (4,6) | 0,83 | 2003-05 | 2010-02 |
| EBR | 2006.28 (2006.4) | 6,16 (5,6) | 0,67 | 2003-05 | 2009-10 |
| TAM | 2005.79 (2005.7) | 8,15 (8,2) | 0,90 | 2003-08 | 2009-10 |
| ASC | 2006.9 (2006.9) | 22,72 (23,4) | 0,91 | 2003-07 | 2010-01 |
| Jerk 2011 | | | | | |
| IMO | $t_0$ (yr) | $A$ (nT/yr) | $R^2$ | initial window | final window |
| NGK | 2011.84 (2011.8) | -6,08 (-6,2) | 0,84 | 2006-08 | 2014-04 |
| EBR | 2009.95 (2011.0) | -7,76 (-6,7) | 0,74 | 2006-02 | 2014-01 |
| TAM | 2009.48 (2009.5) | -6,78 (-6,8) | 0,86 | 2005-10 | 2013-11 |
| ASC | 2010.02 (2012.0) | -19,35 (-19,5) | 0,90 | 2006-12 | 2014-02 |
| Jerk 2014 | | | | | |
| IMO | $t_0$ (yr) | $A$ (nT/yr) | $R^2$ | initial window | final window |
| NGK | 2014.04 (2014.0) | 8.82 (7.2) | 0.68 | 2011-09 | 2014-12 |
| EBR | 2014.02 (2014.0) | 15.28 (12.7) | 0.80 | 2011-06 | 2014-12 |
| TAM | 2013.96 (2014.0) | 13.03 (15.2) | 0.74 | 2011-09 | 2014-12 |
| ASC | 2013.99 (2014.01) | 21.87 (24.9) | 0.79 | 2010-02 | 2014-12 |

## 5 MOSFiT application: determining the influence of the external field filtering methods on geomagnetic jerk detection.

To evaluate the influence of different processing methods, i.e. no data selection and no CHAOS correction (original data), $Kp$ data selection (later labelled KP), IQD data selection (labelled QD) and nighttime data selection (labelled NT) and the CHAOS-7 magnetospheric field reduction (labelled CHAOS), we selected 10 globally distributed INTERMAGNET observatories (Fig. 7) to analyze the 2007, 2011 and 2014 jerks in the X, Y, Z components.

The detection window was individually chosen according to the range of the SV trend for each of the observatories, but always keeping the same for the different processing methods. For each processing method the occurrence time ($t_0$) and the amplitude ($A$) was determined.

In this analysis, only jerks with high values of $R^2$ are considered, that have the typical "V" shape in their SV (note that e.g. in NVS and CNB in 2007 did not show the V-shape) and that were detected by all processing methods. A total of 24 jerks were detected in the Y component, 22 in the Z and only 7 in the X. Note that a total of 12 jerks in the X component were only detectable after using the CHAOS-7 magnetospheric reduction, which shows how important this correction is.



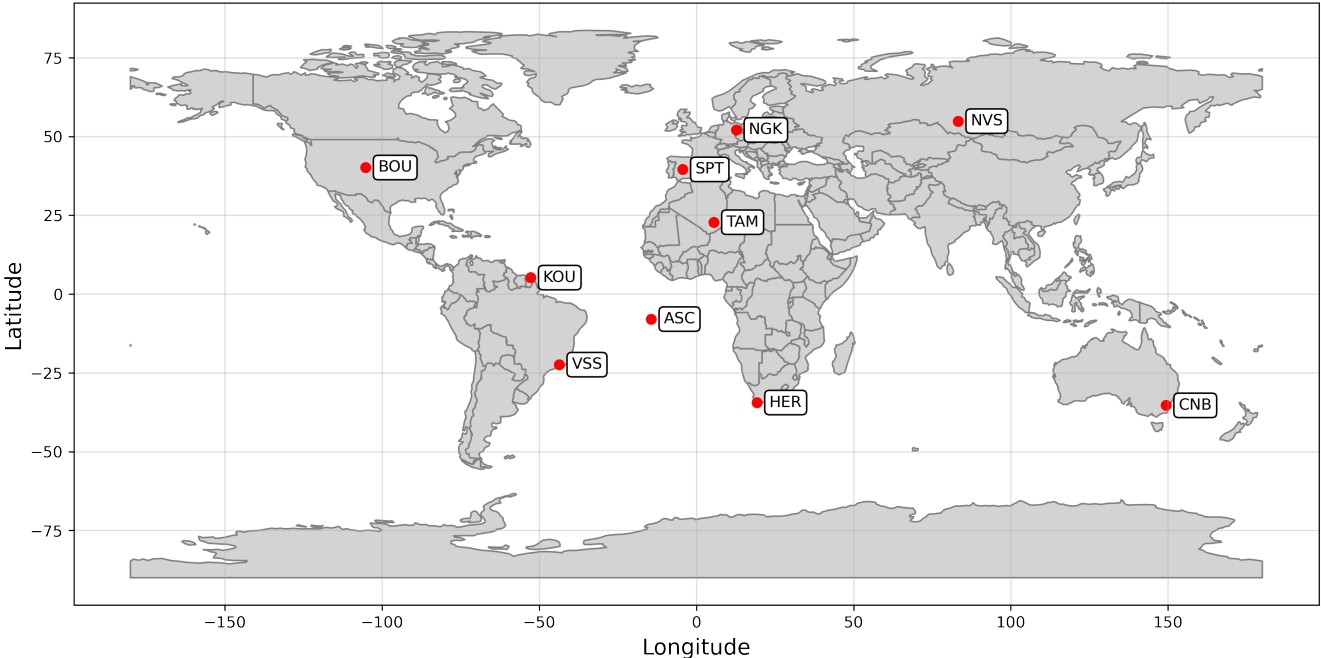

**Figure 7.** INTERMAGNET observatories used in the application for determining the influence of the external field filtering methods on geomagnetic jerk detection. The three letter IAGA code indicates the observatory.

The upper panels in Fig. 8 show the mean of the absolute difference in jerk occurrence time (in days) of the processing methods $Kp$, QD, NT and CHAOS compared to using the original data. The mean occurrence time ($\Delta t_0$) difference between CHAOS correction and original data is largest for the X component (349 days) where it is about 7 times larger than for the processing methods KP, QD and NT (44, 53, 33 days, respectively). A similar pattern is seen for Z, where CHAOS gives 105 days, which is about 5 times larger than KP, QD and NT (23, 19, 22 days, respectively). From this comparison alone one cannot define if the CHAOS magnetospheric correction is either a lot better or a lot worse in these components than the data selection methods tested. However, given that the ring current has a strong amplitude combined with fast temporal changes compared to a jerk and that its signal is not removed by night time or $Kp$-dependent data selection, one can expect that it is the magnetospheric correction that performs best. This can also be seen in Fig. 9 that compares SV for X at Chambon-la-Foret magnetic observatory (CLF) for each processing scheme (blue) with the SV prediction by CHAOS. It clearly shows that the magnetospheric correction by CHAOS gives much better similarity to the CHAOS SV prediction than the three data selection processing methods KP, QD and NT, which all give SV time series that are quite similar to that of the original data with the full content of external fields. Figure 9 shows the application of the different processing methods (KP, QD, NT and CHAOS), including original data, into the Chambon-la-Foret geomagnetic observatory (CLF) X SV calculation, compared to CHAOS-7 model core field. The effectiveness of the processing methods clearly illustrates the results obtained by the investigation.





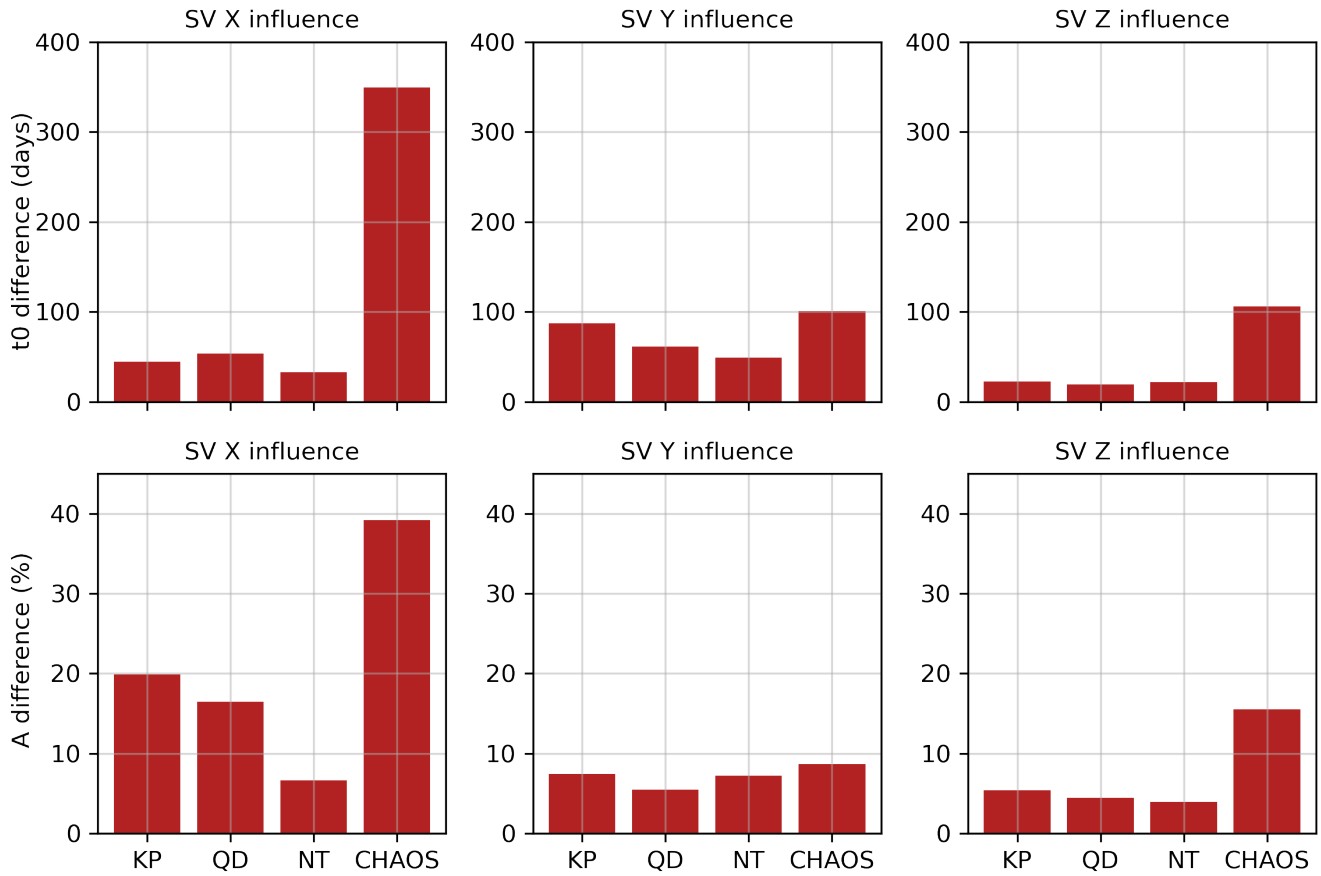

**Figure 8.** Upper panels: mean absolute difference in jerk occurrence time $\Delta t_0$ in days between processed data and original observatory data for different processing methods KP, QD, NT and CHAOS (for explanation see text) for X, Y and Z . Bottom panels: the same as upper panels but for jerk amplitude difference ($\Delta A$) in %

For the Y component (Fig 8, middle upper panel) the absolute shift in ($t_0$) is similar for all processing methods with CHAOS changes ($t_0$) most (by 100 days) and NT least (by 50 days). For the jerk amplitude difference (given in % in the lower panels in Fig 8), we observe a similar pattern as for occurrence time $t_0$.



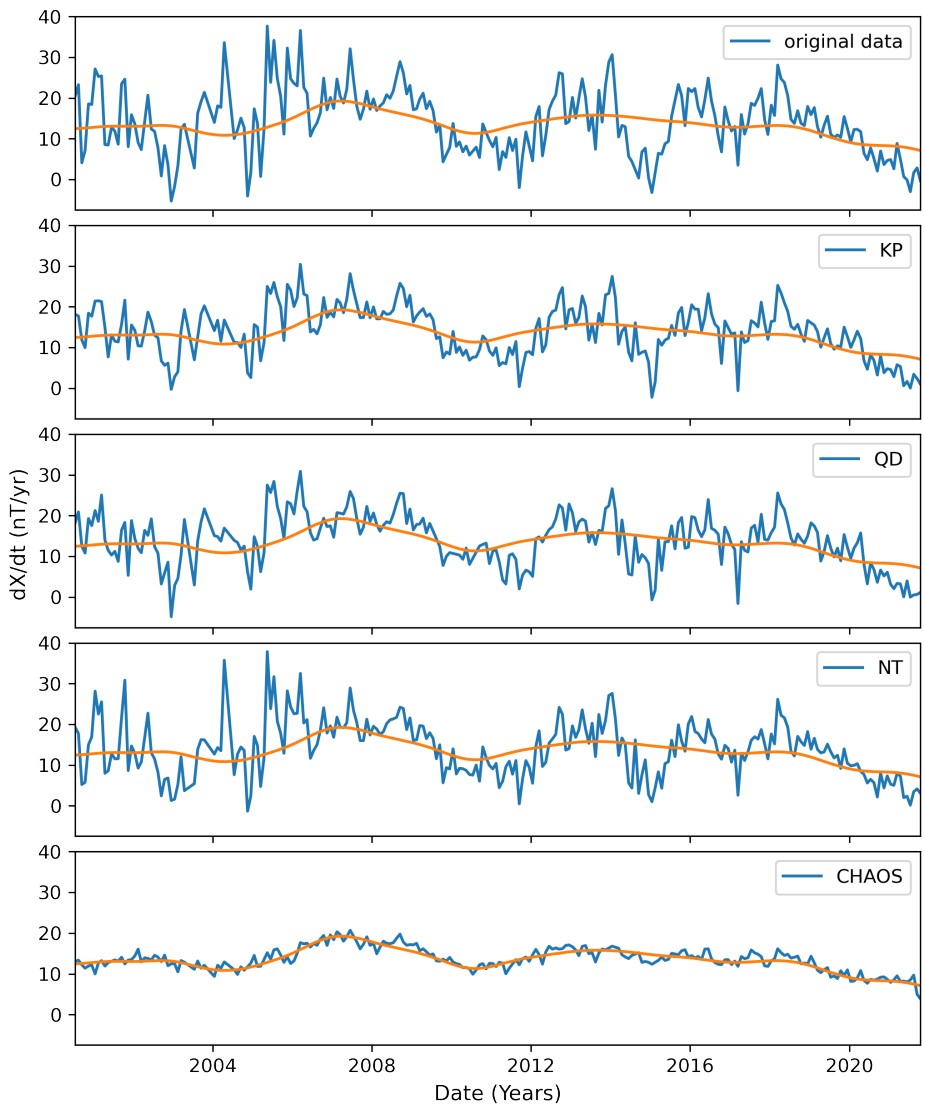

**Figure 9.** Chambon-la-Foret geomagnetic observatory (CLF) X SV calculation using the processing methods KP, QD, NT and CHAOS, and original data, compared to CHAOS-7 core field.

**6 Conclusion**

We present the python package MOSFiT ("Magnetic Observatory and Stations Filtering Tool") to investigate the geomagnetic secular variation (SV) in observatory data. MOSFiT is designed to work with one minute INTERMAGNET definitive and quasi-definitive data. However, it can also be applied to any geomagnetic observatory or magnetometer data. The package offers outlier rejection (Hampel filter), four data selection options (quiet days, disturbed days, $Kp$ index and nighttime period), and



a magnetospheric field reduction using CHAOS-7. MOSFiT can resample the data to hourly, daily, monthly and annual means and provides a method to determine geomagnetic jerk occurrence time ($t_0$) and amplitude ($A$). All steps can be visualized. We successfully validate the implementation of CHAOS in MOSFiT against results for more than 150 observatories presented by Finlay et al. (2020) as well as the MOSFiT geomagnetic jerk detection method against results for three jerks (2007, 2011, 2014) presented by Torta et al. (2015). We investigate the difference between the observatory data SV and the CHAOS core field SV (by calculating its RMS) for uncorrected and observatory data corrected by the CHAOS magnetospheric field prediction for 115 geomagnetic observatories in low, mid and high latitudes. For the uncorrected observatory data, this difference is smallest in the Y component. The magnetospheric field correction reduces this difference most strongly (by about two thirds) in the X component in low and mid latitudes. In general, the differences after the magnetospheric correction are similar for the X, Y and Z components and similar to the difference for the uncorrected Y component. We further quantified the effect of different observatory data processing methods on the determination of the jerk occurrence time ($t_0$) and the jerk amplitude ($A$) for the three jerks for a subset of 10 geomagnetic observatories and found that the CHAOS magnetospheric correction performs best, especially for the X and Z component. MOSFiT is available at https://github.com/marcosv9/MOSFiT-package and we expect it can be a great help for geomagnetic observatory data analysis, allowing convenient and automatic access to the most recent geomagnetic observatory data and indices. In particular, it will allow a timely identification of any future geomagnetic jerks and it can be used as a tool for geomagnetic observatory data quality control.

*Code and data availability.* https://github.com/marcosv9/MOSFiT-package

*Author contributions.* MVS, KP, AO and JM designed the study. MVS did the programming, made the plots and wrote the text. KP and JM contributed to writing, all authors have read the manuscript.

*Competing interests.* We declare no competing interests.

*Acknowledgements.* MVS acknowledges the support of Coordenação de Aperfeiçoamento de Pessoal de Nível Superior (Capes/Brazil, grant 88887.514400/2020-00). The results presented in this paper rely on data collected at magnetic observatories. We thank the national institutes that support them and INTERMAGNET for promoting high standards of magnetic observatory practice (www.intermagnet.org).



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
