# Peer review of "Analysis of geomagnetic observatory data and detection of geomagnetic jerks with the MOSFiT software package"

_EGUsphere, 2023_

## Referee Comment (RC3)

Revision of the paper entitled:

**Analysis of Geomagnetic Observatory Data and Detection of Geomagnetic Jerks with the MOSFiT Software Package**

The basic purpose of MOSFiT, a software package for the presentation and filtering measuring data of Earth's magnetic field, is isolating and analyzing secular variations (SV). It is a tool for researchers working on the models of the Earth's magnetic field or the person who is preparing the data for this purpose. The software package is an important contribution to the development of measuring data processing. During the latest decades the quantity of measurement data has increased enormously.

SV is a time change, calculated as the first-time derivative of the geomagnetic field, calculated on the basis of monthly or yearly means values thereof. The second-time derivations of the geomagnetic field are called geomagnetic jerks (GJ). SV appears as a series of straight-line segments separated by GJ. As the vast majority of the magnetic field originates in the interior of the Earth, this indicates that GJ are of internal origin and their short-time duration that is due to a change in the fluid flow at the surface of the Earth's core.

MOSFiT is also intended to determine the properties of the external magnetic field and control the quality of data about the variation of them, which are measured at an individual observatory or magnetometric station. In addition to data quality control, it also makes possible an easier reestablishment of measuring systems after they have been broken down for various reasons. For this purpose, in addition to the minute mean values, the daily mean values have been calculated additionally. Due to exclusion of extreme values, the median values are a better solution. Even the data, from which the influence of space has been extracted, are useful for research of the local magnetic field as well as the quality of data, measured at an individual observatory or magnetometric station.

Geomagnetic impulses are also the second-time derivations, but they are based on daily acceleration of the Earth's magnetic field, measured at an individual observatory or magnetometric station. They present not only an unpredictable change on graph but also accelerations in the change of the energy density of the geomagnetic field [$J/m^3$]. These accelerations are caused by the conversion of magnetic energy in the upper layers of the Earth. The treatment of the Earth's magnetic field, measured at an individual point on its surface, on the bases of physical quantities expands its research above the comparison of measurement data of geomagnetic field. For a single measurement point on the Earth's surface it is difficult to predict geomagnetic impulses because with today's technical means it is impossible to measure changes in the Earth's interior precisely enough. But tracking changes of energy density of the geomagnetic field has already demonstrated some importance in further research.

Rudi Čop

Portorož (Slovenija), 18. 8.2023

---

## Author Comment (AC2)

**Response letter to the referee comments for manuscript "Analysis of geomagnetic observatory data and detection of geomagnetic jerks with the MOSFiT software package"**

We thank the editor for handling our manuscript and the three referees Seiki Araki, Jan Reda and Rudi Cop for their constructive comments. Below, in red, we respond to the points raised by the reviewers. A revised manuscript with annotated changes is attached (blue: new text, red strike through is deleted text).

We are prepared to publish the current version of the MOSFiT software package with a DOI and link it to our manuscript.

In addition to the changes due to the referee comments, we made changes due to the comment made by J. M. Torta about the following part of the manuscript: **To validate the methods provided in this manuscript, in Sect. 4.2 we compare the automatic detection of the 2007, 2011 and 2014 jerks using observatory data from NGK, EBR, TAM and ASC with that given by Torta et al. (2015). The comparison is presented in Table 4. The results show a very good agreement for the time of occurrence of the jerks as well as for the amplitudes at all four observatories except for the 2011 time of occurrence jerk reported by Torta et al. for ASC, which was 2012.0. However, Fig. 6 of this manuscript, and indeed Fig 1 of Torta et al. (2015), indicate that both the observatory data and the CHAOS model prediction show a jerk around 2010.**

Comment by J. M. Torta:  **After reviewing our results and notes we can confirm that in fact the occurrence found with our method was 2010.2 (which is very close to the result given in this manuscript), but due to a transcription error we wrote 2012.0. We would be grateful if the authors could indicate this (e.g. as a personal communication) in any revisions to their original manuscript.**

From L 203-205 the text "For the jerk in 2011 occurrence time reported by Torta et al. (2015) for ASC is 2012.0, however Fig. 7 shows that both the observatory data and the CHAOS model prediction show a jerk around 2010, which 205 is the same date as detected by MOSFiT." was deleted from the manuscript. The following note was added to Table 4: "In Torta et al. (2015), the occurrence time of the 2011 jerk at ASC is misspelled as 2012.0 but should read 2010.2 (personal communication, J.M. Torta)", the occurrence time was corrected in Table 4.

**Referee comment #1 (Seiki Asari):**

**Basically I find the paper worth publishing, in which a new python tool MOSFiT for magnetic observatory data analysis is introduced. As the source program has been made open, it can be used by observers and researchers widely for different purposes, not just for secular variation analysis but for data processing as well. To improve the manuscript before publishing, I would give a suggestion as below, for which some further computation (but not tough) is needed to supplement their quantitative results. I would like the authors to consider it and make a necessary revision.**

**On top of its useful function for jerk detection, MOSFiT is featured with various methods for external field mitigation. To characterize the different methods more in detail, I suggest that quantitative comparison of SV misfits derived with all those methods be made in Section**

**4.1, where only two cases (CHAOS-7 corrected and uncorrected SVs) are compared in the manuscript. Analyses for KP, QD and NT method, or even for a combination of their use with the CHAOS-7 method, can also be made to complement Table 3 with their outcomes.**

We agree with the reviewer that this is possible with the MOSFiT package. However, we prefer to not include these additional comparisons of methods here for two reasons. Firstly, the purpose of our manuscript is mostly a validation of the code and this is already achieved with the existing comparison. Secondly, as there a numerous possibilities of comparing methods and combinations thereof, we fear that this would substantially increase the length of the manuscript.

**Furthermore, Figure 9 and the text for its explanation in Section 5 may be moved to Section 4.1, as a qualitative illustration to support the quantitative analyses, which can be referenced from Section 5 now dedicated solely for jerk discussion. (Note there is redundancy in L229-231 for describing Fig.9. Some have already been stated previously in L225- 229)**

Figure 9 is moved to section 4.1 (now it is Figure 5) with the explanation adjusted: ' Fig. 5 compares SV for X at Chambon-la-Foret magnetic observatory (CLF) for different processing methods (blue), i.e. Kp selection, quiet days selection, nighttime selection and CHAOS correction, with the SV prediction by CHAOS. It clearly shows that the magnetospheric correction by CHAOS gives much better similarity to the CHAOS SV prediction than data selection processing methods, which all give SV time series that are quite similar to that of the original data with the full content of external fields'

**It is often observed that K-indices are rather small in recovery phase of storms, when the ring currents remain, still lowering X-component level. Therefore I imagine, a combination of the KP-method and CHOAS-7 method (or otherwise a selection with Dst-index) would be even more effective for excluding external fields from data.**

We fully agree with the reviewer that combining a Kp criterion with the CHAOS-7 method would be a very good approach, and that this approach can be easily realized by using the MOSFiT package.

**Typos:**

**L15 variesranges**

Changed to 'ranges'.

**L17 Abrupt changes in SV 'trend' (as described properly in L145)**

Changed from 'Abrupt changes in the SV' to 'Abrupt changes in the SV trend'

**L118 can be can be**

Changed to 'can be'.

**L193 latest the end**

We changed from 'We use only the data until latest the end of 2014' to 'We use only the data until the end of 2014'.

**Referee comment #2 (Jan Reda):**

**GENERAL COMMENTS**

The authors present their experiences related to the analysis of jerks, which are unpredictable events that suddenly speed up secular variations (SV) of the Earth's magnetic field.

Identifying jerks is not an easy task. The authors of this article undertook the difficult task of creating generally available software that performs such a task. The software, called MOSFIT (Magnetic Observatories and Stations Filtering Tool), works with geomagnetic data of Definitive or Quasi-Definitive status in IAGA-2002 format, which are easily available, for example, on the WDC (World Data Center) Edinburgh server. This applies to both INTERMAGNET observatory data and other observatories.

The MOSFIT package can also be used for quality control of data from geomagnetic observatories. The software has filtering, selection, and data visualization capabilities. MOSFIT was written in Python language, which is becoming increasingly popular. The authors provide a link to the software, as well as a guide to facilitate the use of the software package.

In the first chapters, the authors introduce the reader well to the subject of jerks and the problems that arise when detecting them. This is about isolating SV from final recordings of the entire geomagnetic field that also contain the influence of the external field. This is really a big challenge, because not all jerks are very expressive.

In order to isolate jerks, the package offers Hampel filtering as the first step. In the next stage, the contribution of the external field can be reduced by selecting data according to local midnight, Kp geomagnetic indices, quiet days, disturbed days, and by subtracting the magnetospheric field based on the CHAOS-7 model.
It is worth emphasizing that the package offers the possibility of visualization at every stage of work.

The authors presented the detection of jerks from 2007, 2011, and 2014 and compared the results with an analysis performed and published by another researcher. Generally, it can be stated that the consistency of detection was very good, both in terms of the time of occurrence of the jerk and its amplitude.
In summary, I am happy to recommend the manuscript for publication after making minor corrections listed below, in sections SPECIFIC COMMENTS and TECHNICAL, LANGUAGE AND OTHER REMARKS.

**SPECIFIC COMMENTS**

In the bibliography (References), it would be good to add DOI (Digital Object Identifier) identifiers if they are available. DOI identifiers are important because they add credibility to the source and make it easier to access publications. Currently, providing DOI's is basically a standard in bibliographies.

DOIs have been added to references whenever available.

**TECHNICAL, LANGUAGE AND OTHER REMARKS**

**Lines 1,7,44,155,236 The word python should be rather capitalized in this context.**

All occurrences of the word python was changed to Python.

**Line 15 There is missing space between "varies" and "ranges"**

Changed to 'ranges'.

**Line 39 Rather should be "close-to-final" (missing "-")**

Changed to close-to-final.

**Line 58 "computacional" -> "computational"**

Changed to 'computational'.

**Line 126 There is missing space between "time" and "series"**

Changed to 'time series'.

**Lines 174,183 "table" -> "Table"**

Changed to 'Table'.

**Line 232 "Fig 8" -> "Fig. 8" (missing dot)**

Changed to 'Fig. 8'.

**Line 236 Should be rather "one-minute"**

Changed to 'one-minute'.

**Line 282 Change the capitalisation: Journal of the American Statistical Association**

Changed to 'Journal of the American Statistical Association'.

**Line 310 Change the captalisation: Generalized Hampel Filters**

Changed to 'Generalized Hampel Filters'.

**Referee comment #3 (Rudi Čop)**

The basic purpose of MOSFiT, a software package for the presentation and filtering measuring data of Earth's magnetic field, is isolating and analyzing secular variations (SV). It is a tool for researchers working on the models of the Earth's magnetic field or the person who is preparing the data for this purpose. The software package is an important contribution to the development of measuring data processing. During the latest decades the quantity of measurement data has increased enormously.

SV is a time change, calculated as the first-time derivative of the geomagnetic field, calculated on the basis of monthly or yearly means values thereof. The second-time derivations of the geomagnetic field are called geomagnetic jerks (GJ). SV appears as a series of straight-line segments separated by GJ. As the vast majority of the magnetic field originates in the interior of the Earth, this indicates that GJ are of internal origin and their short-time duration that is due to a change in the fluid flow at the surface of the Earth's core.

MOSFiT is also intended to determine the properties of the external magnetic field and control the quality of data about the variation of them, which are measured at an individual observatory or magnetometric station. In addition to data quality control, it also makes possible an easier reestablishment of measuring systems after they have been broken down for various reasons. For this purpose, in addition to the minute mean values, the daily mean values have been calculated additionally. Due to exclusion of extreme values, the median values are a better solution. Even the data, from which the influence of space has been extracted, are useful for research of the local magnetic field as well as the quality of data, measured at an individual observatory or magnetometric station.

Geomagnetic impulses are also the second-time derivations, but they are based on daily acceleration of the Earth's magnetic field, measured at an individual observatory or magnetometric station. They present not only an unpredictable change on graph but also accelerations in the change of the energy density of the geomagnetic field [J/m3 ]. These accelerations are caused by the conversion of magnetic energy in the upper layers of the Earth. The treatment of the Earth's magnetic field, measured at an individual point on its surface, on the bases of physical quantities expands its research above the comparison of measurement data of geomagnetic field. For a single measurement point on the Earth's surface it is difficult to predict geomagnetic impulses because with today's technical means it is impossible to measure changes in the Earth's interior precisely enough. But tracking changes of energy density of the geomagnetic field has already demonstrated some importance in further research.

We are grateful for comments (no changes made to the manuscript).